# Intraoperative Imaging Techniques to Improve Surgical Resection Margins of Oropharyngeal Squamous Cell Cancer: A Comprehensive Review of Current Literature [note 1]

**DOI:** 10.3390/cancers15030896

**Published:** 2023-01-31

**Authors:** Bertram J. de Kleijn, Gijs T. N. Heldens, Jasmijn M. Herruer, Cornelis F. M. Sier, Cesare Piazza, Remco de Bree, Orlando Guntinas-Lichius, Luiz P. Kowalski, Vincent Vander Poorten, Juan P. Rodrigo, Nina Zidar, Cherie-Ann Nathan, Raymond K. Tsang, Pawel Golusinski, Ashok R. Shaha, Alfio Ferlito, Robert P. Takes

**Affiliations:** 1Department of Otorhinolaryngology and Head and Neck Surgery, Radboud University Medical Center, 6225 GA Nijmegen, The Netherlands; 2Department of Oncologic Surgery, Leiden University Medical Center, 2333 ZA Leiden, The Netherlands; 3Unit of Otorhinolaryngology—Head and Neck Surgery, Department of Medical and Surgical Specialties, Radiological Sciences, and Public Health, University of Brescia, 25123 Brescia, Italy; 4Department of Head and Neck Surgical Oncology, University Medical Center Utrecht, 3584 CX Utrecht, The Netherlands; 5Department of Otorhinolaryngology, Jena University Hospital, 07747 Jena, Germany; 6Department of Head and Neck Surgery, University of Sao Paulo Medical School, Sao Paulo 01509, Brazil; 7Department of Head and Neck Surgery and Otorhinolaryngology, AC Camargo Cancer Center, Sao Paulo 01509, Brazil; 8Department of Otorhinolaryngology-Head and Neck Surgery, University Hospitals Leuven, 3000 Leuven, Belgium; 9Department of Oncology, Section Head and Neck Oncology, KU Leuven, 3000 Leuven, Belgium; 10Department of Otolaryngology, Hospital Universitario Central de Asturias, University of Oviedo, ISPA, IUOPA, CIBERONC, 33011 Oviedo, Spain; 11Institute of Pathology, Faculty of Medicine, University of Ljubljana, 1000 Ljubljana, Slovenia; 12Department of Otolaryngology-Head and Neck Surgery, Louisiana State University-Health Shreveport, Shreveport, LA 71101, USA; 13Department of Otolaryngology-Head and Neck Surgery, National University of Singapore, Level 7 Tower Block, National University Hospital, 1E Kent Ridge Road, Singapore 119228, Singapore; 14Department of Otolaryngology and Maxillofacial Surgery, University of Zielona Gora, 65-417 Zielona Gora, Poland; 15Head and Neck Service, Department of Surgery, Memorial Sloan Kettering Cancer Center, New York, NY 10065, USA; 16Coordinator of the International Head and Neck Scientific Group, 35122 Padua, Italy

**Keywords:** intraoperative imaging, narrow band imaging, high resolution microendoscopic imaging, confocal laser endomicroscopy, ultrasound, (auto) fluorescence imaging, augmented reality, transoral surgery, frozen section analysis, computed tomography

## Abstract

**Simple Summary:**

In head and neck cancer, there are several treatment options. When surgical treatment is chosen, removal of the entire tumor is necessary for optimal therapy of the patient. This, however, is challenging in vulnerable areas of the body such as the mouth and throat, as a more radical resection leads to more severe functional limitations after surgery. Several imaging techniques facilitate the distinction of tumor versus adjacent healthy tissue during the operation, which can help the surgeon remove the entire tumor with optimal functional outcomes. In this paper, we aim to provide an overview of these imaging techniques applicable to oropharyngeal squamous cell carcinoma and discuss the possibilities for optimizing the surgical outcome of patients.

**Abstract:**

Inadequate resection margins in head and neck squamous cell carcinoma surgery necessitate adjuvant therapies such as re-resection and radiotherapy with or without chemotherapy and imply increasing morbidity and worse prognosis. On the other hand, taking larger margins by extending the resection also leads to avoidable increased morbidity. Oropharyngeal squamous cell carcinomas (OPSCCs) are often difficult to access; resections are limited by anatomy and functionality and thus carry an increased risk for close or positive margins. Therefore, there is a need to improve intraoperative assessment of resection margins. Several intraoperative techniques are available, but these often lead to prolonged operative time and are only suitable for a subgroup of patients. In recent years, new diagnostic tools have been the subject of investigation. This study reviews the available literature on intraoperative techniques to improve resection margins for OPSCCs. A literature search was performed in Embase, PubMed, and Cochrane. Narrow band imaging (NBI), high-resolution microendoscopic imaging, confocal laser endomicroscopy, frozen section analysis (FSA), ultrasound (US), computed tomography scan (CT), (auto) fluorescence imaging (FI), and augmented reality (AR) have all been used for OPSCC. NBI, FSA, and US are most commonly used and increase the rate of negative margins. Other techniques will become available in the future, of which fluorescence imaging has high potential for use with OPSCC.

## 1. Introduction

Head and neck squamous cell carcinomas (HNSCCs) are the most frequent malignant tumors in a functionally important and vulnerable area of the human body. Smoking and alcohol are important risk factors in developing HNSCC, but viruses also play an important etiologic role in a subgroup of these tumors. In recent decades, the incidence of oropharyngeal squamous cell carcinoma (OPSCC) has increased among younger patients in the Western hemisphere due to the increase of human papillomavirus (HPV)-associated OPSCC and it is expected to keep increasing for the older, unvaccinated population [1,2,3]. Primary treatment of HNSCC consists of surgery and/or (chemo)radiotherapy ((C)RT) [3]. Early stage HNSCC can be treated with a single treatment modality (radiation or surgery), whereas for locally advanced HNSCC a multimodal treatment is often needed. Every modality adds its own nuances and morbidity. Patients who survive HNSCC often suffer lifelong adverse effects, with high morbidity [4,5,6]. Therefore, minimizing the number and extent of treatment modalities (i.e., de-escalation of treatment) without compromising oncological outcomes is crucial for better functional results and consequently improved quality of life.

The choice of the primary treatment of a tumor in a particular site of the head and neck is based on the efficacy of the treatment and the best expected oncological outcome. However, the expected (functional) sequelae and toxicity of each treatment modality as well as the accessibility of the tumor for surgery are other important factors [3]. The treatment of choice in oral cancer is in principle surgery, whereas the treatment of choice for nasopharyngeal cancer is (C)RT [7,8]. The oropharynx is not easily accessible for open surgery, often needing an extensive procedure to remove the tumor. This has been, in general, one of the reasons for a shift from surgery (with postoperative RT) to (C)RT as primary treatment of OPSCC [3,9].

A renewed interest in transoral surgery (TOS) has followed the introduction of minimally invasive techniques such as transoral laser microsurgery, videolaryngoscopic surgery, ultrasonic surgery (TOUS), and robotic surgery (TORS) [10,11]. TORS has gained popularity for the treatment of OPSCC because of its better swallowing outcome and quality of life (QOL) of the patients [12,13,14,15,16]. Compared to RT, the long-term differences in QOL are small or insignificant but swallowing seems better after primary radiotherapy [6,17,18]. The extent of the resection in the larynx and pharynx not only affect survival, but also swallowing and voice function. Therefore, a tailored excision is needed, preserving all physiological functions without compromising local control of the disease. The current standard for intraoperative assessment of resection margins (IOARM) is based on white-light assessment of tumor extension and frozen section analysis (FSA) when available. The definitive histopathology is used for definition of the margin status. The occurrence of involved margins after TORS for OPSCC ranges up to 20% [19,20,21,22,23], although lower rates of positive surgical margins compared to open surgery could be achieved with TORS [24]. Incomplete primary resections result in an increased risk of recurrence and often an escalation of therapy, with postoperative (C)RT adding to morbidity and costs [15,25,26,27].

Reports on the results of TORS show that approximately two-thirds of the patients primarily treated with TORS received additional treatment with (C)RT [28]. To achieve “first time right (initial clear margins)” surgery, i.e., adequate resection without the need for adjuvant or subsequent treatment, IOARM is imperative. 

This review will specifically focus on in vivo and ex vivo techniques for IOARM that can optimize the surgical outcomes of OPSCC. The intraoperative imaging techniques discussed in this review will be narrow band imaging (NBI), (auto) fluorescence imaging (FI), fresh frozen section assessment (FSA), ultrasound (US), confocal laser endoscopy (CLE), high-resolution microendoscopic imaging (HRME), intraoperative computed tomography (CT), and augmented reality (AR). 

## 2. Materials and Methods

A systematic search of the literature was performed using the Embase, PubMed, and Cochrane databases, using the terms “oropharynx”, “cancer”, “margin” and “intraoperative imaging”.

Appendix A shows the search strings per database. The last search was conducted on 7 November 2022. All articles between 1 January 2000 and 7 November 2022 were included. Study selection was performed on titles and abstracts. Only articles that described intraoperative imaging techniques of the oropharynx used for tumor delineation were included. Studies that did not report on OPSCC or OPSCC and oral squamous cell carcinoma (OSCC) separately were excluded, as were articles not available in English and full text. Pre-clinical studies were excluded. Next, a full-text review was performed for final inclusion. The screening was performed by one of the authors (B.J.d.K.). Data extraction was conducted from text and tables. Imaging technique(s), their negative margin definition and sensitivity, specificity, positive predictive value (PPV), and negative predictive value (NPV) were extracted if reported. 

## 3. Results

Through this search strategy, 3972 articles were selected. After title and abstract screening, 40 articles were selected and 19 articles passed the full-text screening. Three more articles were found through reference checks. (See study selection flow chart, Figure 1) Included studies are shown in Table 1. All studies were non-comparative and non-randomized studies. 

### 3.1. Superficial and Deep Margin

#### 3.1.1. Frozen Section Assessment

Intraoperative FSA is an established technique for improving adequate resection margins. It requires close collaboration between the surgeon and pathologist. After excision of the tumor, it is directly assessed for adequate margins using fresh frozen sections. Sampling error is a concern, so the tumor is macroscopically judged by the surgeon and pathologist, after which the pathologist fabricates sections at places where the margin is doubtful. The frozen sections are made directly and assessed by the pathologist. Direct feedback is given to the surgeon so an additional resection can be performed if necessary. A “mapping” system can be used to determine the location of inadequate margins [29]. FSA can be performed in a specimen-driven or defect-driven approach, the first being performed on the excised tumor and the second using biopsies of the surgical bed. 

Seven manuscripts describing the intraoperative use of FSA for OPSCC were found. In a systematic review, Gorphe et al. found that the rates of final positive margins were significantly higher in studies where FSA was not reported [26]. The use of frozen sections for tonsillar tumors has been described by Hinni et al. They used “margin mapping”, inking the specimens in the operating room and sending them for fresh frozen analysis, to achieve 98.5% clear margins on final pathology in a cohort of 100 patients [30]. Herruer et al. used FSA to identify unknown primary tumors in the oropharynx. As the specimens were analyzed intraoperatively, a margin assessment also took place. They found an NPV of 91.8% for margin status on frozen-section pathology in a cohort of 50 patients [31]. In the last two mentioned studies, no differentiation was made between OSCC and OPSCC. Tirelli et al. found a high sensitivity and specificity (94.6% and 94.7%, respectively) when using FSA during piecemeal resection of the tumor [32]. A defect-driven approach was preferred by the authors, because it allowed an easier relocation of the sampling site if a surgical enlargement was required. In contrast, Horwich et al. found that a specimen-oriented protocol resulted in significantly fewer positive margins on final pathology compared to tumor bed sampling [33]. Mackay et al. published a retrospective cohort study with two year follow-up to determine if a specimen-based approach led to better survival rates [34]. They concluded there is no difference between specimen- or defect-based FSA for 2-year survival rates of p16-positive OPSCC. Yu et al. found difficulty in interpretation of FSA in p16-positive OPSCC. The biopsies were often inconclusive (11%) or there were changes from frozen to final pathology (11%) [35]. They found a reference biopsy of the tumor could improve sensitivity from 82.8% to 88.9%. Further data are shown in Table 2.

Although proven to have better outcomes with significantly fewer positive margins, the technique is time-consuming and sensitive to sampling error [26]. Specimen-based sections seem to be superior compared to tumor bed sampling [36]. A frozen–permanent section discrepancy rate of 5.4% in oral an oropharyngeal cancer has been described by Serinelli et al. [37], affecting 16% of 132 patients, but only in 2.5% did it have impact on tumor management. Discrepancies may be ascribed to block sampling, gross sampling, interpretation, or technical error. In conclusion, FSA is an established method for improving resection margins, both superficial and deep, but it is vulnerable to sampling error and inaccuracy in re-resection locations.

#### 3.1.2. (Auto) Fluorescence Imaging

Intraoperative FI is a promising technique for delineating oral and oropharyngeal tumors [38,39,40]. It has also shown potential in the diagnostic workup for unknown primary tumors (at presentation) of the oropharynx [41]. Fluorescence is defined as the spontaneous emission of red-shifted light by a molecule after it absorbs energy from light at a shorter wavelength (higher photon energy) [42]. Two main approaches can be distinguished: autofluorescence by the tissue itself or fluorescence from the use of agents, which can be untargeted or targeted [43].

Autofluorescence such as label-free fluorescence lifetime imaging (FLIm) uses optical contrast created by changes in tissue structure and biochemistry resulting from pathological conditions to differentiate between normal and neoplastic tissue [44]. It is based on endogenous fluorophores in human tissue, such as nicotinamide adenine dinucleotide [NADH], flavin adenine dinucleotide [FAD], and collagen crosslinks in the stroma. Altered metabolism in tumor cells, less collagen, and epithelial thickening in tumor tissue can all change the wavelength and lifetime of fluorescence. Using steady-state (spectral or intensity) autofluorescence imaging, researchers found that measurements were affected by experimental conditions such as non-uniform tissue illumination, irregular tissue surfaces, or blood in the operating field. A downside of autofluorescence as opposed to fluorescence with the use of agents is that deep margin assessment is not possible. Time-resolved autofluorescence techniques resolve the dynamics of the fluorescence decay (lifetime), addressing the limitations of steady-state based methods [45]. 

Fluorescent agents such as indocyanine green (ICG), 5-aminolevulinic acid (5-ALA), and fluorescein isothiocyanate (FITC) can be used to mark the tumor and differentiate it from normal tissue [42]. They are injected and not administered topically as proflavine used in HRME is. Different agents require specific wavelengths of light, such as near infrared (NIR), to detect. Generally, the penetration depth of light into the body from the surface depends on the wavelength of the light and the presence of absorbing substances such as hemoglobin and water in the body [46]. This leads to different characteristics of the images depending on the agents used. NIR imaging using ICG can achieve higher tissue penetration and almost an absence of autofluorescence [46]. Untargeted fluorophores such as ICG rely on perfusion and accumulation of the agent in tumor tissue due to leaky capillaries. This means the differentiation between tumor and healthy tissues is qualitative and not quantitative. One important problem is the rapid clearance of these fluorophores, reducing the operating time window. 5-ALA-based fluorescence is more tumor-specific, depending on the accumulation/activation of a non-fluorescent substrate in highly proliferating tumor cells, and requires a blue light spectrum, leading to lesser depth of penetration compared to NIR light [46]. EGFR- and VEGF-targeted antibodies, which are equipped with an NIR fluorescent dye, are seen as promising tracers, but have not been specifically investigated in OPSCC [47]. 

Three manuscripts were found describing the intraoperative use of FI for OPSCC. Gorpas et al. demonstrated a setup in which multispectral time-resolved fluorescence spectroscopy (ms-TRFS) was used during TORS for oral cavity surgery. The goal of this study was to determine usability for oropharyngeal tumors. The integrated system was designed to collect and analyze ms-TRFS data in real time from areas of interest in the oropharynx prior to and after surgical excision of cancer [48]. In practice, a fiberoptic wire is inserted through an EndoWrist introducer of the TORS system; this wire is connected to the ms-TRFS console. Measuring at different points and storing the data on an imaging computer, a heatmap of the tumor can be created and projected over the normal WL image seen by the surgeon to aid in differentiation between tumor and healthy tissue. Measurements are performed in real time. The imaging depth is <400 μm [48]. From the same group, Weyers et al. showed a statistically significant change (*p* < 0.001) between tumor and healthy tissue in at least one FLIm parameter in all in vivo pre-resection and eight out of nine ex vivo post-resection assessments. The same research cohort showed a statistically significant change in FLIm parameters between tumor and healthy tissue when used with a combination of machine learning and visualization by a classification heat map (tumor and healthy tissue) [45]. Marsden et al. continued the exploration of FLIm with a study to demonstrate the diagnostic ability of FLIm as a means of intraoperative guidance during OPSCC surgery. They used a freehand scanning approach to scan 53 patients with oral and oropharyngeal neoplasms [39]. A statistically significant change between healthy tissue and cancer was observed in vivo for the acquired FLIm signal parameters linked with metabolic activity, demonstrating the potential of FLIm for reliable intraoperative margin assessment.

Limitations of FI are dependent on the technique used. To improve the sensitivity of fluorescence techniques, different labeled fluorescent agents are available for topical or intravenous use, and new tracers are in development, with promising results [49]. The use of these techniques may improve surgical margin detection, although specific evidence is not yet available for oropharyngeal lesions. Apart from in vivo use of fluorescence, ex vivo use may also prove to identify close margins of resection and identify areas for further resection or investigation by FSA or other techniques [47]. Weyers et al. have built a custom FLIm instrument to perform their studies on autofluorescence but widespread usage of such an imaging technique in oropharyngeal surgery is not available [45]. Autofluorescence is not suitable for deep margin assessment. In conclusion, FI can be used to delineate the superficial and deep margins of neoplastic tissue both in vivo during resection (real time) and ex vivo after resection. Although promising, the use of (auto)fluorescence for OPSCC is still in development, and not readily available in clinical practice.

### 3.2. Superficial Margin

#### 3.2.1. Narrow Band Imaging

One of the most established non-invasive imaging techniques used in the upper aerodigestive tract is NBI. NBI is an endoscopic optical imaging enhancement technology that uses a filter to narrow white light (WL) into two 30 nm bands of blue and green light (415 and 540 nm, respectively) simultaneously. The blue light visualizes superficial mucosal vascular patterns, whereas the green light shows small changes in submucosal microvasculature to a depth of 0.24 mm [50,51]. The use of NBI for detection of well-demarcated areas and irregular superficial microvascular patterns associated with upper aerodigestive tract dysplasia and squamous cell carcinoma has been established [52,53,54,55]. Higher sensitivity and specificity have been found for the use of NBI compared to white light (WL) for HNSCC [56]. In 2010, Muto et al. performed a multicenter, prospective, randomized controlled trial and reported significantly higher sensitivity, NPV, and accuracy with NBI compared to WL for the detection of HNSCC, including OPSCC [57]. Using NBI in endoscopy, oropharyngeal lesions can be classified by assessing modifications of the intrapapillary capillary loops, making differentiation of malignant lesions more sensitive compared to using WL inspection alone [54,58].

Four manuscripts described the intraoperative use of NBI for OPSCC. Two prospective studies were by Tirelli et al., who have published one article on OPSCC alone and several articles on OPSCC and OSCC combined. In 2016, these authors reported on NBI in OPSCC, leading to a statistically significant reduction in the rate of positive superficial margins in definitive histopathology from 36.4% to 11.5% (*p* = 0.028) [59]. In 2018, the same group published prediction accuracy rates (sensitivity, specificity, PPV, and NPV) for involved mucosa using NBI in OPSCC (Table 2) [60]. Further data are shown in Table 2. The use of NBI combined with TORS for resection of OPSCC has been described by Tateya et al. in a case report, in which they reported to have successfully determined the extent of the resection using NBI [61]. Azam et al. have shown the use of deep learning on NBI images of OPSCC to achieve high levels of precision and accuracy [62].

NBI is readily available and implemented in modern imaging processors. The learning curve for the use of NBI is steep [57,63]; to address this, machine learning or artificial intelligence (AI) can be used to aid physicians in detecting aberrant vascularization in the mucosa using NBI, with encouraging results [64,65]. Use of AI technology has been established for differentiating between benign and malignant polyps in colonoscopies [66]. The technology shows promising potential for future use in the oral cavity and oropharynx [67,68]. NBI does have its limitations: it is only suitable for determining superficial margins as its two bandwidths are absorbed by the hemoglobin in the superficial layers of the mucosa, thus the penetration of the NBI light (however greater than that demonstrated by WL) is limited to the upper layers of the epithelium and mucosa (180–250 μm) due to the soft tissue absorption phenomenon [69]. Deep resection margins cannot be assessed using NBI. It is also essential to have a blood-free mucosal area to assess since blood will always appear black through the filter, consequently blinding the view [56,70]. Whereas some authors reported that mucosa that was previously treated with (C)RT undergoes changes in vasculature, making detection of new or recurrent HNSCC with NBI more difficult [56,71], other research groups found no influence of previous therapy on the use of NBI to optimize margin control [63,72]. Further investigation is needed to determine the usability of NBI after (C)RT [52,54]. The combined use of NBI and magnifying endoscopy may further improve margin assessment in a post-(C)RT setting, but further investigation is needed [52,54]. 

The descriptions of the IPCL changes or dysplastic epithelium were originally described in the squamous mucosa of the esophagus and hypopharynx. These are carcinogen-induced cancers that transform from a dysplasia to in situ carcinoma to invasive cancer. HPV-related oropharyngeal carcinoma, however, does not progress the same as carcinogen-induced cancers. The premalignant phase of HPV-related oropharyngeal cancer is not well described and the IPCL changes of HPV-related oropharyngeal cancers have not been well studied. Further studies to delineate the microvascular changes of the epithelium in HPV-related oropharyngeal cancer is required before NBI can be reliably employed for margin demarcation in HPV-related oropharyngeal cancers. In conclusion, NBI seems to be a useful complementary aid during transoral oropharyngeal surgery, allowing for real-time assessment of mucosa and leading to fewer positive superficial margins. 

#### 3.2.2. Confocal Laser Endomicroscopy

CLE was developed to obtain a high magnification of the mucosa of the gastrointestinal tract [73], where it has found its way into routine clinical use [74]. CLE achieves a 1000× magnification to visualize intercellular spaces in vivo, allowing for a high-resolution and microanatomical analysis of tissue (“optical biopsies”) in real time during endoscopy [75]. CLE uses a low-power laser to illuminate tissue and subsequent detection of the light reflected from the tissue is used to form high-resolution cellular imaging and evaluation of tissue architecture during endoscopy. Topical or intravenous fluorescence contrast agents can be used to improve resolution [76]. Several minutes after administering fluorescein intravenously, image quality deteriorates [77]. In practice, a small rigid probe can be inserted transorally, or the technique may be implemented in the endoscope itself [73]. The depth of scanning ranges between 0–250 μm [75] for the abovementioned soft tissue absorption phenomenon [69]. CLE use in HNSCC has yielded promising results, but the true value of this method is still to be determined [74].

The literature search to find articles describing the intraoperative use of CLE for OPSCC yielded two articles. Sievert et al. published a feasibility study of intraoperative assessment of safe margins with CLE on five patients. They concluded that CLE could contribute to a more precise evaluation of surgical cancer margins [78]. The inter-rater reliability was low, with a κ-value of 0.60. Sensitivity, specificity, PPV, and NPV to detect involved mucosa are shown in Table 2. Dittberner et al. showed high SN, SP, PPV, and NPV and accuracy for CLE matched with final pathology in a pilot study on 13 patients (Table 2) [79]. This study showed a mixed group with 52.9% OPSCC. Data on oropharyngeal cancer was partially shown separately, stating that 83.1% to 98.6% of CLE images were fitting to the histopathology.

Limitations of CLE are the limited field of view, depth of penetration, difficulty of interpretation of the image for oropharyngeal lesions, and inability to assess the deep margin [80]. The most-reported optical probe provides images of 240 μm, making it difficult to assess the whole tumor margin [80]. When used by an experienced head and neck surgeon, low specificity and high sensitivity were found in OPSCC, possibly because the oropharyngeal mucosa has an atypical aspect compared to oral mucosa [78]. In studies published on the use of CLE on HNSCC, only the mucosal margins were assessed and the feasibility of deep margins was not evaluated [80]. Difficult-to-assess sites form an obstacle for the use of CLE, as the probe has to be placed on the mucosal area. Ex vivo use of CLE on the resected tumor specimen has also been described, circumventing this problem but somewhat reducing its practical application and meaning [81]. CLE is also associated with considerable expense, as a scanning unit costs around $200,000 and a single application $275 ($250 for one probe use and $25 for the contrast agent fluorescein 10% 5 mL), with a decrease in image quality when recording takes too long [78]. Inter-observer differences are common and the technique has an intermediate learning curve [78]. Machine learning can be used to aid the physician in interpreting images made with CLE [82]. Further research will be needed to determine the place of CLE in the diagnosis of oropharyngeal neoplasms and intraoperative margin assessment. In conclusion, the use of CLE may improve superficial resection margins, but inter-rater reliability is low and cost is relatively high. Low specificity is found in oropharyngeal lesions. 

#### 3.2.3. High-Resolution Microendoscopic Imaging

HRME can be used to assess the superficial mucosal margin of a tumor on a cellular level. As the technique can be used intraoperatively by the surgeon, FSA by a pathologist may be surpassed. It is a noninvasive technique utilizing a fiberoptic probe and topical fluorescent nuclear contrast agent (i.e., proflavine hemisulfate 0.01%) to obtain images of epithelial architecture and cellular morphology [83]. Proflavine hemisulfate is the hemisulfate salt form of proflavine, an acridine-derived fluorescent contrast and disinfectant agent. The probe consists of a coherent bundle of optical fibers, allowing for a 720 μm circular field of vision and a depth of roughly 50 μm, displaying at 12 frames per second [83,84]. The small field of view makes it useful for examination of suspicious areas, but not the whole excision margin. It is a low-cost technique that has a relatively short learning curve when the physician has a cytological and histopathological background. It can achieve high levels of sensitivity and specificity, discriminating between benign and neoplastic mucosa in the head and neck [85,86].

Our search yielded one article on HRME for OPSCC. In 2015, Patsias et al. reported the use of HRME for OPSCC treated with TORS. As this was a feasibility study with three patients included, there were no results on the efficacy of HRME, but they did show that it is possible to provide a real-time assessment of superficial mucosa in the oropharynx and added value in oncologic surgery, whereas the assessment of deep margins using HRME is an active area of ongoing research [84].

HRME is a relatively low-cost endoscopic solution for in vivo margin assessment on a cellular level, costing <$5000 [87,88]. Limitations of this technique are the risk of misinterpretation if the quality of the image is unsatisfactory, the use of topical contrast agents, and the problem of deep margin assessment due to limited depth penetration. The interpretation of HRME images is dependent on the number of nuclei in the image. Algorithms can be used to improve the PPV and NPV of HRME [89]. The strong affinity of proflavine contrast agent for keratin can mask the underlying nuclei in heavily keratinized tissue, thus limiting the ability to interpret the images obtained. Additionally, imaging of the deep muscle margins with this technology remains to be proven effective [84]. In conclusion, HRME is feasible for use in OPSCC but further research is needed. The technique is not commercially available.

### 3.3. Deep Margin

#### 3.3.1. Ultrasound (US)

US is a dynamic imaging technique, using soundwaves in the megahertz range. It is noninvasive and quick [90]. The use of intraoperative US has been described in detail in treating oral tongue carcinoma. A recent systematic review including 19 articles states that the technique can be used safely in oral cavity carcinoma, with tumor thickness measurements correlating well with histopathology [91]. This has also been confirmed by Filauro et al., demonstrating the favorable performance and cost-effectiveness ratio of intraoral US when compared to both MR and final histopathology [92]. Free margin status of OSCCs can be improved by using intraoperative US. In addition, frequency of positive margin status (<1 mm) can be significantly reduced (5% vs. 15%, *p* < 0.001) [93].

We found two papers dealing with the intraoperative use of US for OPSCC. Clayburgh et al. used an US transducer in TORS to identify large-caliber vessels and to take measurements of tumor thickness to determine deep margin. The tumor thickness measurement was found to be accurate within 1 to 2 mm compared to gross measurement of the tumor thickness after resection [94]. Pazdrowski et al. used intraoperative US in 20 cases of tonsillar cancer. They concluded that intraoperative US is a safe, non-invasive method for improving surgical margins [95].

As the oropharynx is a confined space, it is challenging to use instruments such as US probes. The use of tUS on ex vivo resection specimens has been reported by Noorlag et al. [96], de Koning et al. [93,97], and Brouwer de Koning et al. [98]. They found that not only the ex vivo specimen is assessable for tumor depth by US but also the tumor-free margins. This latter application will improve applicability of US in OPSCC. In conclusion, the use of US has been described in OPSCC with or without TORS. It can be used to determine deep tumor margins in vivo and in real time, but data on efficiency, sensitivity, specificity, PPV, and NPV are not available. Ex vivo analysis may be applicable for OPSCC.

#### 3.3.2. Computed Tomography and Magnetic Resonance Imaging

Preoperative imaging such as CT and magnetic resonance imaging (MRI) provides important information about the location of tumors, arteries, and other vital structures needed for safe and accurate TOS. Intraoperative CT has been used in endonasal surgery, cochlear implantation, and soft tissue surgery [99]. However, while in intraoperative position, the soft tissue becomes displaced compared to the preoperative position due to neck extension, laryngoscopes, and retractors. Intraoperative imaging could be used to correct for this displacement. 

Our search yielded three articles on the use of intraoperative CT for improvement of resection margins. The use of an “intraoperative” CT on cadaver heads based on image-guided navigation has been demonstrated in two studies. Kahng et al. showed the time needed to find beads placed in the oropharynx was shorter and the accuracy of finding them was higher with the use of “intraoperative” CT scanning [100]. Ma et al. demonstrated considerable displacement of Teflon beads inside the oropharynx and carotid arteries between pre- and intraoperative positions on CT, with a significant reduction in target registration error when using the tracker linked to the intraoperative scan [101]. An in vivo proof of concept study performed by Paydarfar et al. showed a 1 mm registration accuracy during TOS within the pharynx and larynx using electromagnetic trackers for target localization [102]. For imaging, an intraoperative contrast enhanced CT and external fiducials were used in combination with a CT-compatible suspension laryngoscope. Tracking was performed using an optical tracking technique. 

The cost of implementation of a CT or MRI in the OR is one of the major concerns of this technique. After that, limitations are mainly dependent on the static image obtained by the scanner. Intraoperative changes in patient positioning or soft tissue mobilization change positioning toward the CT scan, which is static. This can be overcome by making a new scan after repositioning or by using real-time imaging techniques such as US in combination with CT [102]. Ex vivo use of the scanners may aid in quickly assessing margins after resection, as reported in tongue cancers [103,104]. In conclusion, intraoperative use of CT or MRI for OPSCC shows promising results, but further research is needed to determine clinical relevance. To use this technique there is a need for a center with CT/MR scanner at the operation rooms.

## 4. Discussion

### 4.1. Resection Margins and Adequate Margins

To determine a free, close, or involved margin in HNSCC, generally a macroscopically tumor-free margin of >5 mm, <5 but >1 mm, and <1 mm, respectively, is used, but this is dependent on different subsites. For example, in OSCC 5 mm is often regarded as an adequate margin leading to better OS [105,106,107]. In glottic laryngeal cancer, much smaller margins are accepted, particularly in transoral microscopic laser surgery, as they do not lead to worse OS or disease-free survival [108]. Which margin should be considered as an adequate resection margin for OPSCC is still point of intense investigation. In this review, different articles used different values for adequate margins (Table 1). Different cutoff values have been suggested in the literature, both wider and smaller than 5 mm [27,30,109,110]. The location of the tumor may also influence an acceptable margin. For example, in tonsil or pharyngeal wall tumors smaller margins may be accepted if anatomical barriers are uninvolved, whereas tongue base tumors where no anatomical barriers are present require bigger margins. In addition, HPV-positive status influences the acceptable margin, as the etiology of these tumors is different from non-HPV-related HNSCC, and response to therapy is better [111,112]. However, HPV-positive tumors in smokers behave differently and may require more aggressive treatment compared to non-smokers [113].

Several studies on TORS for OPSCC use different definitions between 2–5 mm for R0 resections (Table 3). 

The general consensus is that involved and close margins lead to worse OS and more local recurrence. Tumor characteristics such as perineural growth or growth pattern also play an important role in the risk of local recurrence and OS, leading to more indications for adjuvant treatments [27,105]. However, the trend seen in the literature of accepting margins of <5 mm for OPSCC, will also lead to a reduction of adjuvant treatment, reducing toxicity of the overall treatment.

Independent of the definitions of adequate margins, techniques for IOARM can be evaluated for their capacity to obtain predefined desired distance from the tumor to the surgical margin. What the surgeon will do with accurate information on the resection margin distance in clinical decision making is another discussion.

### 4.2. Most-Applicable Techniques for Oropharynx

Advantages and disadvantages of all the above-described techniques have been summarized in Table 4. 

As resections of oropharyngeal carcinoma are mainly affected in the deep margin [107], there is a need for techniques to improve deep margin control. Most in vivo techniques, however, are aimed at the superficial margin. In vivo assessment of the superficial margin can be performed using NBI, HRME, CLE, or FI. For in vivo deep margin assessment US, FI, and CT/MRI can be used. Ex vivo margin assessment has been performed for a long time using FSA. More research is needed, but US, CT, MRI, HRME, CLE, FI, or FSA may also be suitable for ex vivo margin assessment. 

The confined space and different mucosal aspects of the oropharynx complicate the use and interpretation of imaging techniques. NBI and FSA are the most frequently used and tested intraoperative techniques to improve adequate surgical margins for the superficial (NBI) and deep (FSA) surgical margins. Tirelli et al. reported an exploratory study combining NBI with TORS piecemeal resection and FSA, facilitating both superficial and deep resection margin assessment. Results seem promising, although the evidence is still limited [32]. 

Both NBI and FSA are very usable in the oropharynx and are readily available. NIR ICG fluorescence and CLE are also licensed but less commonly used. NBI has shown to be a valuable aid, consuming little time with high sensitivity, specificity, PPV, and NPV. Other techniques for assessment of the superficial margin in the oropharynx such as HRME are still under development and have to prove they are easily accessible and better. Yang et al. have shown an imaging system integrating WL, autofluorescence (AF), and HRME. The main goal of this system is to differentiate between benign and neoplastic lesions in the oral cavity [119]. The system is based on locating suspicious areas with WL and AF with high sensitivity and exploring the region with HRME to improve specificity. A comparable system may be of use in the oropharynx to detect suspicious areas and to determine the superficial resection margin. Further research will be needed to determine the place of HRME in the diagnosis of oropharyngeal neoplasms and intraoperative margin assessment.

FSA is useful for the deep margin of OPSCC but it is time consuming and has risk of sampling error. Furthermore, this is an ex vivo appreciation of the specimen or of biopsies of the defect, leading to relocation errors if an extra resection has to take place. A numbering system may be used to determine the exact location of the inadequate margin on the specimen that is traceable to the defect [29], but a real-time assessment during the resection would be preferable. US is easily accessible in many clinics. It has been shown to improve resection margins in vivo, but is less usable in the confined space of the oropharynx. Ex vivo use of the US is also viable, but may lead to the same relocation bias as FSA. In the future, AF shows promise to be used for both superficial and deep margin assessment. The use of lifetime AF has shown great potential [39], although a “multi bloc” resection may be needed to determine the deep margin intraoperatively.

### 4.3. Future Developments

In vivo biopsy by Raman spectrometry (RS) has a high discriminatory value between tumor and healthy surrounding tissue [120,121]. It relies on the scattering of monochromatic light, which interacts with molecular vibrations affecting up-shifting or down-shifting in the energy of photons and provides an objective, nondestructive, and fast intraoperative assessment of the resection surface, including the deep soft tissue layers [122,123]. RS does not need a contrast medium. In practice, this technique uses a (fiber) optic probe, which may be used superficially or inserted in the tissue with a needle in order to detect the shift in wavelength of photon scattering by different tissues [120]. No publication was found on the use of RS for OPSCC and the technique is still in the early stages of development for medical use, but it may be applicable for in vivo and ex vivo analysis of resection margins in the future. The combination of anti-Stokes Raman scattering, two-photon excited fluorescence, and second harmonic generation microscopy in a multimodal nonlinear microscopy setup in combination with automated image analysis on ex vivo slides has been shown to have good results in HNSCC and shows potential if it can be implemented in vivo [124].

Optical coherence tomography (OCT) works analogous to US, using reflecting coherent light to make cross-sectional images of the underlying tissue in real time, with a depth of 0.5–2 mm on a microscopic level [43]. No reports have been found of use in the oropharynx, but a combination of optical coherence tomography, Raman spectroscopy, and fluorescence lifetime microscopy has been described for ex vivo use on head and neck tumors [125].

The performance of reflectance-based hyperspectral imaging (HSI) on HNSCC has been described by Halicek et al. [77]. HSI is an imaging technique based on the scattering of light due to inhomogeneity of biological structures and dissimilar absorption of different components of tissue. The absorption, fluorescence, and scattering characteristics change in tumors compared to normal tissue. These differences can be captured by HSI in spatially resolved spectra, providing diagnostic information about the tissue [126]. Halicek found HSI and autofluorescence to be more sensitive than proflavine and red–green–blue images for SCC detection. In an ROC analysis, the area under the curve of oropoharyngeal cancer was 0.95 (with 2 mm margin) and 0.91 in HPV-positive SCC in tonsillar tissue. They concluded that HSI and autofluorescence imaging can accurately detect the cancer margin in ex vivo specimens within minutes.

HSI is not fluorescence imaging, but instead uses white light. It was found that autofluorescence was more sensitive than proflavine and red–green–blue images for SCC detection. An area under the curve for oropharyngeal cancer was 0.95 (with 2 mm margin) and in HPV-positive SCC in tonsillar tissue 0.91. It was concluded that HSI and autofluorescence imaging can accurately detect the cancer margin in ex vivo specimens within minutes [127].

Augmented reality (AR) can be combined with several techniques to improve tumor delineation. With the use of TORS, the surgeon is looking through lenses at a projected image. This image can be digitally enhanced with imaging to digitally project the tumor in the surgical field. To apply AR intraoperatively, a 3D reconstruction or 2D image of an MRI or CT scan can be created and matched with the positioning of the patient intraoperatively using osseous landmarks or applied markers [128]. A challenge in the use of pre-operative imaging is the mobility of the individual structures of the neck. The anatomical proportions of an awake patient with a closed mouth differ considerably from a sleeping patient in hyperextension with an oral retractor. Therefore, preoperative imaging cannot be accurately registered to an intraoperative situation. The use of intraoperative imaging can improve the level of registration accuracy [102]. We did find several articles on AR used on oropharyngeal cancer, but all were pre-clinical feasibility studies [129,130,131]. An overview of AR during OPSCC surgery is described in more detail by Pratt et al. [128]. The overlay of PET/CT and MRI is shown as proof of concept. Chan et al. have shown the use of preoperative CT or MRI in combination with AR to provide information on surrounding (vascular) structures during tumor resection [132]. The use of conventional imaging may prove to be less applicable during the resection of large tumors as the operational field will change during the procedure. To overcome the use of static imaging in a dynamic situation, a combination of AF lifetime imaging and AR during TORS can be used [48].

As previously stated, the results of fluorescence may improve with the use of tumor-specific fluorescence agents. Recent research has shown potential for the use of fluorescent-labeled panitumumab and cetuximab. Cetuximab-IRDye800CW has also shown great potential for use in HNSCC [133]. An alternative is the use of panitumumab. Panitumumab is an EGFR antibody that can be combined with NIR dye (e.g., IRDye800CW) to label HNSSCC in vivo [134]. Another promising tool for molecular margin investigation is an injected fluorescence agent coupled to PARRP1 [135,136]. Further research is needed to determine the most relevant technique(s).

## 5. Conclusions

To achieve margin improvement in OPSCC, NBI and FSA are now established techniques for superficial and deep margin assessment, respectively. However, there is a need for additional real-time in vivo assessment of the deep layer. FI and US have shown great potential to macroscopically aid the surgeon during resection of the deep margin while suspicious sites can be checked using CLE or, in the future, HRME. In the future, a combination of imaging techniques is likely to provide the most optimal (near) real-time information on resection margins, combining the benefits of different techniques.

## Figures and Tables

**Figure 1 cancers-15-00896-f001:**
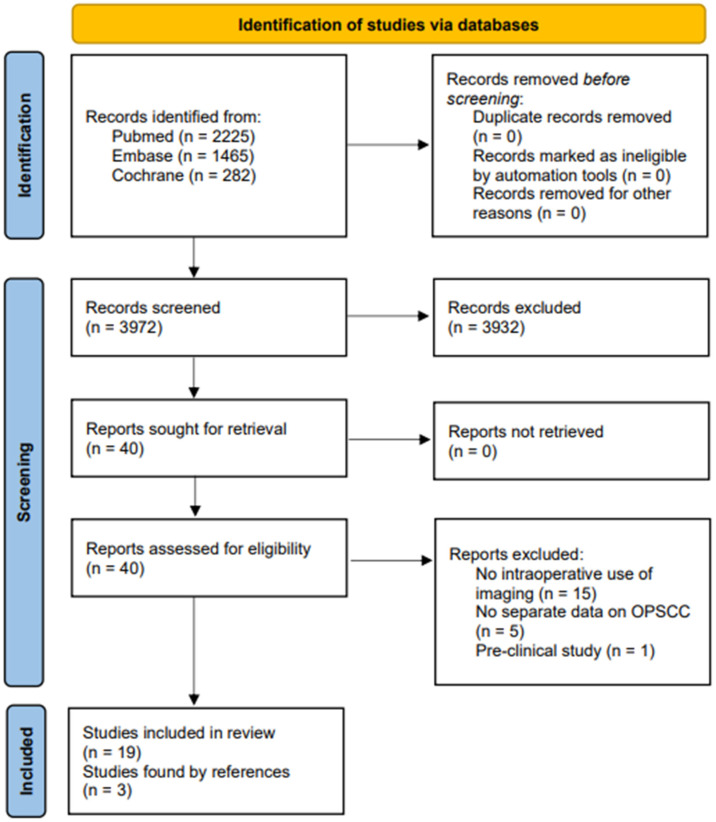
Study selection flow chart.

**Table 1 cancers-15-00896-t001:** Included articles.

First Author	Year	Technique Used	Control Group	Study Design	T-Stage	N = (Intervention) OPSCC	N = (Control) OPSCC	Surgical Procedure	Application of Intervention	Negative Margin Definition	Close Margin Definition	Superficial or Deep Margin
Gorphe P	2019	FS	No FS	SR MA	Nd	2547	1367	Nd	AR	No tumor in margin	NA	Nd
Hinni M	2013	FS	NA	Nd	1–4	128	NA	TO	AR	No tumor in margin	NA	Superficial and deep
Herruer J	2020	FS	NA	Nd	1–3	50	NA	TO	AR	>1 mm	Nd	Nd
Tirelli G	2019	FS	NA	Nd	1–4	80	NA	TO	AR	>3 mm	Nd	Deep
Horwich P	2021	FS	NA	Nd	1–4	108	NA	TO	AR	<5 mm	1–5 mm	Nd
Mackay C	2022	FS	Na	Nd	1–2	90	NA	TO	AR	Nd	Nd	Nd
Yu A	2022	FS	Na	Nd	Nd	170	NA	TORS	AR	Nd	Nd	Nd
Gorpas D	2019	Fl AR	NA	DR	Nd	4	NA	TORS	BR DR	Nd	Nd	Superficial and deep
Weyers B	2019	Fl	NA	Nd	Nd	10	NA	TORS	BR AR	Nd	Nd	Superficial and deep
Marsden M	2021	Fl AR	NA	Nd	Nd	50	NA	TORS	BR AR	Nd	Nd	Superficial and deep
Tirelli G	2016	NBI	WL	CT	1–4a	14	14	TO	BR	>3 mm	0.1–3 mm	Superficial
Tirelli G	2018	NBI	WL	Nd	1–4	22	NA	TO	BR	>3 mm	0.1–3 mm	Superficial
Tateya I	2014	NBI	WL	CR	T1	1	NA	TORS	BR	Nd	Nd	Superficial
Azam M	2022	NBI	NA	Nd	NA	NA	NA	NA	NA	NA	NA	NA
Sievert M	2021	CLE	NA	Pi	2–3	5	NA	TO	BR	Nd	Nd	Superficial
Dittberner A	2021	CLE	NA	Nd	Nd	13	NA	ND	BR	Nd	Nd	Superficial
Patsias A	2015	HRME	NA	CR	Nd	3	NA	TORS	DR	Nd	Nd	Superficial
Clayburgh D	2016	US	NA	Nd	Nd	10	NA	TORS	BR	Nd	Nd	Deep
Pazdrowski J	2010	US	NA	Nd	1–4	20	NA	Nd	BR	Nd	Nd	Deep
Kahng P	2019	CT	NA	Pi	NA	4	NA	NA	NA	NA	NA	NA
Ma A	2017	CT	NA	Pi	NA	4	NA	NA	NA	NA	NA	NA
Parydarfar J	2019	CT	NA	Nd	2	1	NA	TO	BR	Nd	Nd	Nd

Technique used: CLE = confocal laser endoscopy, CT = computed tomography, Fl = fluorescence, FS = frozen section, HRME = high-resolution microscopic endoscopy, NBI = narrow band imaging, US = ultrasound. Study design: CR = case report, CT = controlled trial, NA = not applicable, Nd = not defined, Pi = pilot, SR MA = systematic review with meta-analysis. Surgical procedure: TO = transoral, TORS = transoral robotic surgery, LP = lateral pharyngotomy, TM = transmandibular. Application of intervention: BR = before resection, DR = during resection, AR = after resection.

**Table 2 cancers-15-00896-t002:** Summary of articles organized by technique.

Technique	Author	Summary	Remarks
FSA			
	Gorphe et al. 2019	Eight series reported systematic frozen section analysis. The cumulative number of patients was 501, of whom 25 (5%) had positive final margins. Sixteen series reported on-demand frozen section analysis, depending on the intraoperative assessment of the quality of the resection. The cumulative number of patients was 2046, of whom 69 (4.6%) had positive final margins. Thirteen series did not report frozen section analysis, with a cumulative number of patients of 1367, of whom 169 (12.3%) had positive final margins. The chi-squared comparison test was significant (*p* < 0.0001).	
	Hinni et al. 2013	There was one positive margin encountered in the previously untreated group (1%) and one local recurrence ultimately developed.	With mean follow-up of 4.3 years (range, 2–14 years), the 5-year estimate for local control was 99%, disease-free survival was 94.5%, and overall survival was 76%.
	Herruer et al. 2020	Intraoperative frozen section margin assessment has shown potential, with a specificity of 92% compared to final histopathology.	Of the 50 intraoperatively found tumors, 98% (n = 49) had negative margins on frozen sections, and 90% (n = 45) were truly negative on final histopathology. Eighteen patients (29.5%) avoided adjuvant treatment.
	Tirelli et al. 2019	Piecemeal resection of the tumor using TLM was performed. After resection, margin mapping was performed by taking superficial margins of the mucosa around the tumor and deep margins by taking bowls of tissue underlying the site of the resection. Comparison between frozen section and definitive histological examination found a sensitivity, specificity, PPV, and NPV of 93.6%, 96.8%, 90.7%, and 96.8%, respectively.	In both groups, tissue to be analyzed on frozen section was collected from the tumor bed because a defect-driven approach was preferred. The whole deep margin was examined in 2–3 slices.
	Horwich et al. 2021	Implementation of a specimen-oriented frozen section protocol resulted in 1 of 111 patients (0.9%) having positive final pathology margins, a statistically significant decrease (*p* < 0.001).	Recurrence-free survival at 3 years was 88.4 and 50.7% for negative and positive final margins, respectively (*p* = 0.048).
	Mackay et al. 2022	Two-year OS for patients in the defect study arm was 100% (SE, 0%; 95% CI, 100–100%; n = 17), while for patients in the specimen study arm, it was 97% (SE, 2.2%; 95% CI, 93.8–100%; n = 49; *p* = 0.6). Two-year DSS for both study arms was 100%, with a standard error of 0% (*p* > 0.99); two-year local control rates for defect and specimen sampling were 100% (SE, 0%; 95% CI, 100–100%; n = 17) and 98% (SE, 2.1%; 95% CI, 94.1–100%; n = 49), respectively. Lastly, 2-year recurrence-free survival in the defect and specimen arms was 94.1% (SE, 6.1%; 95% CI, 83.6–100%; n = 17) and 95.8% (SE, 3%; 95% CI, 90.2–100%; n = 49; *p* = 0.29), respectively.	Data on p16+ OPSCC were presented separately; 90 patients with OPSCC were included. T1-2 and N0-2a.
	Yu et al. 2022	The diagnostic value of intraoperative frozen margin analysis was evaluated. Overall accuracy was noted to be 94.1%, with sensitivity of 85.1%, specificity of 97.4%, positive likelihood ratio of 32.7, and negative likelihood ratio of 0.15. Positive margin controls improved sensitivity from 82.8% to 88.9%. It also improved diagnostic utility of a positive intraoperative margin, as the positive likelihood ratio increased from 29.6 to 37.0 (difference, 7.4 [95% CI, 5.0–9.8])	A total of 170 patients were included in this retrospective study.
Fl			
	Gorpas et al. 2019	Time-resolved fluorescence spectroscopy (TRFS) was used to complement the visual inspection of oral cancers during transoral robotic surgery (TORS) in real-time and without the need for exogenous contrast agents. Label-free and real-time assessment and visualization of biochemical tissue features during the robotic surgery procedure has the potential to improve intraoperative decision making during TORS.	A prototype TRFS instrument was integrated synergistically with the da Vinci surgical robot and the combined system was validated in swine and human patients.
	Wyers et al. 2019	In vivo region-level discrimination reached a sensitivity of 86% and specificity of 87% using the Random Forests (ensemble learning) method. FLIm parameters of dysplasia were analyzed separately and were found to be between the parameters of tumor and healthy tissue.	
	Marsden et al. 2021	FLIm point measurements acquired from 53 patients (n = 67,893 pre-resection in vivo, n = 89,695 post-resection ex vivo) undergoing oral or oropharyngeal cancer removal surgery were used for analysis. Statistically significant change (*p* < 0.01) between healthy and cancerous tissue was observed in vivo for the acquired samples.	The developed approach demonstrates the potential of FLIm for fast, reliable intraoperative margin assessment without the need for contrast agents. No differentiation was made between oral and oropharyngeal measurements.
NBI			
	Tirelli et al. 2016	The use of NBI on OPSCC led to a statistically significant reduction in the rate of positive superficial margins observed from 36.4% to 11.5% (*p* = 0.028) in definitive histology. The use of NBI increased the resection margin, with a mean of 11 ± 3 mm, consequently leading to a resection margin of 25 ± 4 mm from the macroscopic tumor edge in certain areas.	In this study, the NBI group was compared to a historic cohort comparable for tumor and size, although there were more early-stage tumors in the historic cohort.
	Tirelli et al. 2018	The use of NBI allowed for a more precise definition of tumor superficial extension in 70.5% of the patients. The sensitivity, specificity, PPV, and NPV of NBI in OPSCC were 85.7% [ 57.2–98.2], 75% [34.9–96.8], 85.7% [57.2–98.2], and 75% [34.9–96.8], respectively.	The use of NBI was not influenced by tumor site in oral and oropharyngeal SCC.
	Tateya et al. 2014	Single case report using magnifying endoscopy with NBI intraoperatively on OPSCC of the tongue base using TORS.	
	Azam et al. 2022	With a model of SegMENT + ensemble TL and a backbone of Xception, intersection over union of 0.784, a dice similarity coefficient of 0.879, a recall of 0.907, a precision of 0.919, and accuracy of 0.933 were achieved.	Data on oropharyngeal carcinoma presented separately.
CLE			
	Sievert et al. 2021	Tumor margin was examined with CLE and biopsy during tumor resection. We calculated an accuracy, sensitivity, specificity, PPV, and NPV of 86%, 90%, 79%, 88%, and 82%, respectively.	Five patients were included. A total of 12.809 CLE frames were correlated with pathology. Inter-rater κ-value of 0.60. IV contrast was used. The examination added 10 min of operation time.
	Dittberner et al. 2021	The concordance between histopathology and CLE images varied between the patients from 83.1 to 98.6% for oropharynx. Further analyses were on a mixed group. The sensitivity, specificity, and accuracy in detecting cancer using the classified CLE images was 87.5, 80.0, and 84.6%, respectively. The positive and negative predictive values were 87.0 and 80.0%, respectively. The procedure would add 9 min of operation time.	Pilot study in 13 patients. Mixed group with oropharynx (52.9%), followed by oral cavity (35.3), and hypopharynx (11.8%) cancers. Data for oropharynx partially shown separately.
HRME			
	Patsias et al. 2015	Three patients were included. The length of the procedure was 4–7 min. HRME images obtained during surgery showed features that were consistent with histologic assessment	
US			
	Claybourgh et al. 2016	Ultrasound used for tumor margin detection in four cases. All margins were free of tumor. Large vessels could also be detected. The use of ultrasound added 5–10 min to operating time.	As there is no dedicated system for use in TORS, a neuro spine or liver transducer was used. One or more robotic arms had to be removed during surgery to allow access for the transducer in the oropharynx.
	Pazdrowski et al. 2010	It was found in this study that intraoperative ultrasonographic examination allows accurate visualization of the tumor mass.	No data were gathered on improvement of resection margins.
CT			
	Kahng et al. 2019	Intraoperative imaging significantly improved localization accuracy and task efficiency when targeting submucosal beads in cadaver heads during operative laryngoscopy.	The imaging was performed on cadavers with beads in the oropharynx to register displacement pre- and post- “operatively”.
	Ma et al. 2017	The purpose of this study was to develop and validate an accurate image-guidance system for TORS. A significant reduction in target registration error was observed when registering the tracker to the intraoperative compared to the preoperative scan.	The imaging was performed on cadavers with beads in the oropharynx to register displacement pre- and post- “operatively”.
	Parydarfar et al. 2019	Suspension laryngoscopy was performed with a CT-compatible laryngoscope on four patients. An intraoperative contrast-enhanced CT scan was obtained and registered to fiducials placed on the neck, face, and laryngoscope. For surgical navigation during TOS, a high level of registration accuracy can be achieved by utilizing intra-operative imaging.	Setup time for the four included patients was long (average 76 min). Tissue displacement during surgery is a limitation when using static imaging such as CT.

Abbreviations of techniques: AR = augmented reality, CLE = confocal laser endoscopy, CT = computed tomography, Fl = fluorescence, FS = frozen section, HRME = high-resolution microscopic endoscopy, NBI = narrow band imaging, US = ultrasound. Summary shows most relevant findings. Abbreviations: TORS = trans oral robotic surgery, PPV = positive predictive value, NPV = negative predictive value, OPSCC = oropharyngeal squamous cell carcinoma, OSCC = oral squamous cell carcinoma, IV = intravenous, TLM = transoral laser microsurgery.

**Table 3 cancers-15-00896-t003:** Definition of R0 margins used in different studies on oropharyngeal cancer.

Study	Clear Margin	p16 Status
EORTC 1420 [114]	>3 mm mucosal margin (Deep R0 is no constrictor invasion)	p16 + and −
ECOG E3311 [111]	>3 mm	p16 +
AVOID [115]	>2 mm	p16 +
ORATOR [18]	>2 mm	p16 + and −
ORATOR2 [116]	>3 mm	p16 +
PATHOS [117]	>5 mm	p16 +
University of Pennsylvania [118]	>2 mm	Not defined

**Table 4 cancers-15-00896-t004:** Clinical usability of all imaging techniques for intraoperative margin assessment in oropharyngeal squamous cell carcinoma.

Imaging Technique	Pros (+) and Cons (−) for Intraoperative Margin Assessment
Frozen section analysis	+ histological confirmation of resection margin − small parts of specimen are screened for involvement − location bias for additional resection − time-consuming − no real-time in vivo assessment
Autofluorescence imaging	+ real-time in vivo assessment + suitable for large areas + does not require fluorescent agents − not suitable for deep margin assessment
Fluorescence imaging	+ real-time in vivo assessment + suitable for deep margin assessment + suitable for large areas + assessment during resection + deep tissue penetration possible with near-infrared − requires fluorescent agents
Narrow band imaging	+ real-time in vivo assessment + no contrast agents needed + suitable for large areas + readily available − assessment needs to be performed pre-excision − not suitable for deep margin assessment
Confocal laser endoscopy	+ real-time in vivo assessment of margin − limited field of view − limited depth of penetration − inability to assess the deep margin − expensive
High-resolution microendoscopic imaging	+ real-time in vivo assessment of margin + relatively inexpensive − deep margin assessment is to be proven −only superficial tissue penetration − not readily available − topical contrast required
Ultrasound	+ real-time in vivo assessment + deep margin assessment + readily available + assessment during resection − difficult to use in hard-to-reach areas
CT/MRI	+ deep margin assessment − no real-time assessment − static imaging − costly to implement in the operating room

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
