# Peer review of "Intraoperative Imaging Techniques to Improve Surgical Resection Margins of Oropharyngeal Squamous Cell Cancer: A Comprehensive Review of Current Literatureâ€"

_cancers, 2023, doi:10.3390/cancers15030896_

Round 1
Reviewer 1 Report
I have just a few minor grammar/unclear phrases
line 236: It appears to be a "run on" sentance
Line 331: I am having touble following all the "cut-offs"
line 387: "which is may be used"??
line 401: I assume you are reffer to a ROC analysis when you state "area under the curve"... Should be more explicit
Author Response
Thank you very much for your positive review and suggestions for improvement. We have used them to revise the manuscript accordingly and hope this will be satisfactory.
With kind regards,
Sicerely,
Robert Takes

Reviewer 2 Report
This is a comprehensive review, summarizing the available evidence regarding a wide range of techniques and approaches aiming to intraoperatively assess resection margins in oropharyngeal carcinomas (including frozen section analysis, autofluorescence imaging, fluorescence imaging, NBI, confocal laser endoscopy, high resolution microendoscopic imaging, ultrasound, intraoperative CT).
As it is well-known, one of the most relevant issues regarding the interpretation and systematization of literature regarding surgical margins in oropharyngeal carcinomas is their definition, which is still heterogeneous. However, Authors adequately addressed this issue, also providing a useful table.
Overall, this article is well written, comprehensive, detailed, and methodologically sonud, providing potentially useful data on approaches which are currently matter of increasing scientific interest, although most of them are still not part of the routine clinical practice.
I'd just recommend to add a PRISMA flow-chart to the methods section, to make more evident to the reader the study selection process.
Author Response
Thank you very much for your positive review and suggestion for improvement. The suggested flow-chart is added to the manuscript.
With kind regards,
Sincerely,
Robert Takes

Reviewer 3 Report
The literature review on intraoperative imaging techniques to improve surgical resection margins of oropharyngeal squamous cell cancer is very comprehensive.
Author Response
Thank you very much for your positive review. We made improvements in the manuscript according to the suggestions of the reviewers.
With kind regards,
Sincerely,
Robert Takes
